# HPLC-DAD-ESI-MS Analysis for Simultaneous Quantitation of Phenolics in Taiwan Elderberry and Its Anti-Glycation Activity

**DOI:** 10.3390/molecules24213861

**Published:** 2019-10-26

**Authors:** Ho-Shin Huang, Hsu-Sheng Yu, Chia-Hung Yen, Ean-Tun Liaw

**Affiliations:** 1R&D Center, King Herb BioMed., Tainan 71201, Taiwan; adinol.huang@gmail.com; 2Department of Food Science, National Pingtung University of Science & Technology, Pingtung 91201, Taiwan; hsyu@mail.npust.edu.tw; 3Department of Biological Science and Technology, National Pingtung University of Science and Technology, Pingtung 91201, Taiwan; chyen0326@mail.npust.edu.tw

**Keywords:** *Sambucus formosana*, anti-glycation, phenolics

## Abstract

*Sambucus formosana* is most commonly used as a traditional herb medicine in Taiwan. In this study, high performance liquid chromatography equipped with photodiode array detection-mass (HPLC–DAD-ESI-MS) method was developed for the identification and quantification of bioactive phenolics. The developed method was also validated for accuracy, precision, limit of detection, and quantification. In this method, chlorogenic acid, rutin, isoquercetrin, nictoflorin, astragalin, and quercetin were quantified in linearity range of 10–100 (μg/mL) with a correlation coefficient of greater than 0.996. High recovery (86.5–93.1%) and good reproducibility were obtained for six phenolics with the relative standard deviation ranging from 1.7–3.1%. Therefore, the proposed method for simultaneous quantification of six bioactive phenolics in the extract and fractions of *S. formosana* using HPLC–DAD-ESI-MS detection under the optimized conditions is accurate and validated. Among the results, methanol extract showed the greatest values of total phenolic content (93.1 mg gallic acid equivalent/g). Additionally, the methanol extract revealed best antioxidant capacity based on the DPPH scavenging activity and anti-glycation activity (IC_50_ was observed at 97.1 and 77.9 μg/mL, respectively).

## 1. Introduction

The Sambucus is a genus of flowering plants in the family Caprifoliaceae. The various species are commonly called elderberry. The Sambucus genus of plants is widely distributed throughout Europe, Asia, and North America, and some of them are well known as medicinal herbs. As reported previously, the Sambucus genus mainly contains polyphenols and anthocyanins [1,2]. European elderberries are known as a traditional remedy for various kinds of diseases, and used to treat common cold symptoms. The healing properties of elderberry are associated with phenolic compounds and scavenging free radicals. The antibacterial, antioxidant activities and medicinal potential of the European elderberry have been demonstrated [3].

Taiwanese elderberries, *Sambucus formosana* and *Sambucus chinensis*, as synonyms belonging to Caprifoliaceae, named in Chinese PA GU XIAO, are native to southern Taiwan and southern China. *S. formosana* is a plant whose recorded history is as a blood circulation invigorating herb, and it is applied externally to treat trauma, infectious wounds, and inflammations by Taiwanese aborigines [4]. *S. formosana* also has various environmental benefits. It can attract pollinators, like butterflies and bees, and provide food for wildlife (Figure 1). Some publications have described chemical research on *S. chinensis*, including lignans, terpenoids, triterpenoids, and phenolic acids, and other compounds were identified by nuclear magnetic resonance [5,6]. Additionally, some components of *S. chinensis* have been analyzed by HPLC [7]. However, to the best of our knowledge, there are no published reports in literature about component analysis by HPLC-MS, antioxidant activity, and anti-glycation of phenolics from *S. formosana.* Thus, developing an efficient and validated method for analyzing phenolics in *S. formosana* is necessary. In this study, a simple and efficient HPLC-DAD-ESI-MS method is proposed for the quantification of the major compounds and also the anti-glycation and antioxidant potential were determined for these plant extracts.

## 2. Results and Discussion

### 2.1. Analysis of Polyphenolic Compounds by High-Performance Liquid Chromatography–Diode Array Detection–Electro-Spray Ionization Mass Spectrometry (HPLC-DAD-ESI (+)-MS)

A HPLC-DAD-ESI-MS method was carried out for the qualitative and quantitative determination of the polyphenolic compounds from *S. formosana*. To achieve good resolution with short analysis time, the mobile phase was optimized through comparisons of different HPLC conditions. Preliminary experiments were conducted to test the chromatographic column. The RP-C18 column (YMC 1.9 μm, C18, 100 × 3 mm) was selected for HPLC-DAD-ESI-MS analysis based on its good separation ability and short analytical time. Under these HPLC conditions and MS optimized conditions described in Section 3.4., the phenolics exhibit stronger signal responses in negative ion mode. Table 1 lists the retention time and MS data for the six standard phenolics.

The six compounds showed excellent linearity with R^2^ > 0.996 and results are shown in Table 2. The limits of detection (LOD) of six compounds were 1.4, 1.0, 1.3, 1.2, 1.5, and 0.8 μg/mL for chlorogenic acid, rutin, isoquercetrin, nictoflorin, astragalin, quercetin respectively, based on a signal-to-noise ratio of 3:1. A high recovery (86.5–93.1%) and precision were acceptable with RSD values ranging between 1.7% and 3.1% for intra-day variation. This validated method was successfully applied to the quality control of *S. formosana* extract and fractions, which provided particularly important information for production and application.

A previous study revealed in the *Sambucus ebulus* berry extract hyperoside (1.60 mg/100 g FW) and quercetin-3-Orutinoside (1.20 mg/100 g FW) were the major polyphenolic compounds identified.

In additional, Kaempferol was found in fewer amounts (0.23 mg/100 g FW) in *Sambucus nigra* crude-extract comparing to *Sambucus ebulus* (0.57 mg/100 g FW) sample [8]. In this study, the HPLC-DAD chromatograms of the methanol extract from *S. formosana* are shown in Figure 2. The quantitative analytical results (Table 3) indicate their contents distribution in these samples. The contents of chlorogenic acid, rutin, isoquercetrin, nictoflorin, astragalin, and quercetin in the extract and fractions were in the range of 0.1–3.5 mg/g. In this results, quercetin exhibited the highest abundance (3.5 mg/g) among six phenolic compounds were simultaneously obtained. The content of bioactive components was affected by different solvent and extraction conditions. To ensure the stability, safety, and efficacy for clinical use, guidelines and quality control for *S. formosana* preparations should be standardized in the future.

### 2.2. The Content of Total Polyphenols and Antioxidant Activity

In previous studies, the solvents such as methanol and ethyl acetate are found superior in concentrating the phenolics, and the recovery of polyphenols from plant materials is influenced by their solubility in the extraction solvent, the type of solvent, the interaction of phenols with other herb constituents, and the formation of insoluble complexes [9]. In this study, the total phenolic content of the *S. formosana* extract and its fractions was expressed as gallic acid equivalent. Among the fractions of *S. formosana*, the ethyl acetate and the n-butanol fractions had the highest and lowest amount of total phenolic content, with values of 69.5 and 7.5 mg Gallic Acid Equivalent (GAE)/g, respectively. The total phenolic content of the methanol extract was 93.1 mg GAE/g (Table 4). Quantitative determination of total phenolic contents indicates that methanol extract possesses the highest concentration of phenolic content [10].

The radical scavenging activities of the *S. formosana* extract and fractions were evaluated using DPPH (2,2-Diphenyl-1-Picrylhydrazyl) radical scavenging assays. The methanol extract and ethyl acetate fraction of *S. formosana* reduced the DPPH significantly (Table 4). The methanol extract exhibited the highest DPPH scavenging with IC_50_ value (IC50 value is the concentration of the sample required to inhibit 50% of radical, 97.1 ± 0.2 μg/mL) as compared to the other fractions of *S. formosanum*. The ethyl acetate fraction, n-butanol fraction, and residual aqueous fraction had IC_50_ values of 120.5 ± 0.5, 520.2 ± 1.5, and 590.7 ± 0.4 μg/mL, respectively. The residual aqueous fraction showed the lowest antioxidant activity with IC_50_ value. Phenolic compounds, widely distributed in herb plants and berry fruits, act as antioxidants or free radical scavengers and have a wide range of health-promoting benefit [11]. Positive correlation between phenolics and DPPH radical scavenging activity was reported previously [12]. The results of our study were similar to a previous study, which revealed that methanolic extract and EtOAc fraction from *S. formosana* had the effective capacity of scavenging for superoxide radicals and correlated with total phenolic content (*R*^2^ = 0.95; *p* < 0.05), thus indicating its antioxidant potential.

### 2.3. Anti-Glycation Activity

Antiglycation activity of the different solvent soluble fractions of *S. formosana* is presented in Table 5. The Methanolic extract and EtOAc fraction exhibited strong advanced glycation end-products (AGEs) formation-inhibitory activity, with IC_50_ values of 77.9 and 99.6 µg/mL compared to the positive control aminoguanidine with the IC_50_ value of 185.5 µg/mL. On the other hand, *n*-BuOH and residual aqueous fraction fractions did not exhibit anti-glycation activity within the test concentrations. A previous study revealed that phenolic and polyphenolic compounds could affect the glycation process [13]. In addition, a previous study illustrated that phenolics might serve as one of the potent AGE inhibitors for the prevention of ageing and diabetes. The phenolics like chlorogenic acid and caffeic acid from fruit wines were recognized as their key α-Glucosidase inhibitory activity [14]. In the present study, the methanol extract of *S. formosana* demonstrated stronger anti-glycation activity which might be attributed to the abundant polyphenols compounds (chlorogenic acid, rutin, isoquercetrin, nictoflorin, astragalin, quercetin). Thus, the results indicate that an appropriate solvent is important for anti-glycation activity. In the future, *S. formosana* could be used as nutraceutical in the treatment of diabetes.

## 3. Materials and Methods

### 3.1. Reagents

HPLC-grade methanol, acetonitrile, and chemicals were purchased from Merck, Mumbai, India. Reference standards of chlorogenic acid, rutin, isoquercetrin, nictoflorin, astragalin, quercetin were purchased from Sigma-Aldrich (St Louis, MO, USA). The chemical structures of the six constituents were listed in Figure 2.

### 3.2. Plant Collection and Identification

The whole plant of *Sambucus formosana* was collected in March 2019 from Tainan herbal market, Taiwan. The plant material was botanically identified by Dr. Sheng-Zehn, Yang, from the Department of Forest at National Pingtung University of Science & Technology. A voucher specimen was deposited in the laboratory of the functional food of National Pingtung University of Science & Technology and the plant code is 201903SF01. Later the stems, leaves, and roots of *S. formosana* were washed and cleaned, then dried in shade at 25–30°. They were weighed, ground in a mechanical grinder, placed in airtight bottles, and stored in the desiccators to be used later for extraction.

### 3.3. Preparation of Extract and Fractions

The plant materials were washed with water and air-dried at room temperature for 3 days. The collected sample was then oven-dried at 50 °C for 3 days and ground to a fine powder by an electronic blender. The dried *S. formosana* (1.5 kg) was mechanically ground to a fine powder and then sieved through a 10 mesh sieve. The obtained powder was extracted with methanol at room temperature. The combined methanol (MeOH) extracts (60 g) were successively partitioned between ethyl acetate (EtOAc, 3 g) and H_2_O. The latter fraction was repartitioned between n-butanol (*n*-BuOH, 6.5 g) and residual aqueous fraction (45.2 g).

### 3.4. HPLC–DAD-ESI-MS Analysis of the Extract

The bioactive compounds present in the methanol extract and different fractions were identified by LC-ESI-MS/MS analysis using a mass spectrometer with Nexera X2 system (Shimadzu 8045, Kyoto, Japan). The *Sambucus formosana* extract samples dissolved in methanol were separated, which was carried out on RP-C18 column (YMC 1.9 μm, C18, 100 × 3 mm). The composition of the mobile phase was a mixture of 0.1% formic acid/H_2_O (solvent A) and 0.1% formic acid/acetonitrile (solvent B). A gradient elution was performed at the flow rate of 0.25 mL/min with the following run conditions: 85% A from 0 to 5 min, 65–45% A during 3 to 8 min, 45–0% A during 8 to 15 min. The column temperate was fixed at 40 °C and 2 μL of the sample was injected and identification of the compounds was done under the conditions of negative ion mode; mass spectra were recorded in the range 100–800 *m*/*z*. Ion spray voltage was at 3.5 KV, interface temperature was kept at 400 °C, and nebulizing gas flow was 20 L/min. The extract and different fractions of *S. formosana* were analyzed by HPLC-DAD-ESI-MS. The identification of flavonoids in *S. formosana* was completed by comparing the retention time and MS spectra with standards. The peak area of each component in the extract and fractions of *S. formosana* was acquired from its chromatogram and the abundance of each compound was calculated from its corresponding calibration curve. Experiments were conducted in triplicate and the resulting data were expressed as the mean ± SE and the unit was represented as mg/g.

The method was validated for linearity, accurate, precision, limits of detection (LOD), and quantification (LOQ). Standards at the concentration range of 10–100 (μg/mL) were prepared. Solutions containing four standards at five different concentrations were injected in triplicate. Linear regression equations were constructed by establishing calibration graph with the peak area (y), concentration (x, μg/mL). The mixed standards solution was further diluted to a certain concentration to explore the limits of detection (LOD) and quantification (LOQ). The LOD and LOQ were determined at a signal-to-noise ratio of 3 and 10, respectively. The intra- and inter-day precisions were determined by continuously injecting the sample solution for three replicates on the same day and by measuring it once a day for three consecutive days, respectively. The recovery test for reflecting accuracy was conducted by the standard addition approach. The recovery yield was carried out according to the following formula: recovery yield (%) = [(amount detected − original amount)/amount spiked] × 100%, and RSD (%) = (SD/mean) × 100%. The repeatability was estimated on the grounds of relative standard deviation (RSD).

### 3.5. Determination of Total Polyphenol Content

The total phenolic content (TPC) of the extract was determined using Folin–Ciocalteu reagent [15]. In brief, a solution of extract (0.5 mL) with proper dilution was mixed with 1.0 mL of Folin-Ciocalteu reagent at room temperature (5 min) and added 2 mL of 7% Na_2_CO_3_ solution. The mixture was boiled for 1 min, and absorbance of the colour was recorded at 750 nm in spectrophotometer (Shimadzu, Kyoto, Japan). The results were expressed in g gallic acid equivalent (GAE)/g, on dry weight basis.

### 3.6. DPPH Radical Scavenging Activity

The scavenging activity of the extract against DPPH (2,2 diphenyl-1-picrylhydrazyl) radical was assessed according to a reference method with some modifications [16]. In brief, 0.1 mL of extract was mixed with 3.0 mL of DPPH solution (0.1 mM). The reaction mixture was left in the dark at room temperature for 30 min. The absorbance of the mixture was measured at 517 nm. The percentage of scavenging activity against the DPPH radical was calculated by the following equation:Radical scavenging activity (%) = (Abs_control_-Abs_sample_)/Abs_control_(1)
where, Abs control is the absorbance of DPPH radical in methanol; Abs sample is the absorbance of DPPH radical solution mixed with sample extract. IC_50_ value indicating the concentration at which a sample would inhibit free radicals by 50% was also calculated. All determinations were performed in triplicate (*n* = 5).

### 3.7. In Vitro Anti-Glycation Assay

Advanced glycation end-products (AGEs) were produced in vitro using a method described previously [17]. In brief, Bovine Serum Albumin (BSA, 20 mg/mL) in phosphate buffered-saline solution (PBS, pH 7.4) containing 0.02% sodium azide was incubated with glucose (500 mM) at 37 °C for 0, 7, 14, 21, and 28 days in the absence (control) and presence of each of the extracts (100, 200, and 400 μg/mL) or AG (200 μg/mL). Each solution was kept in the dark in a capped vial, and incubation was allowed to proceed in triplicated vials. For time course experiments involving fluorescent AGE formation, we measured the fluorescence (370-nm excitation wavelength and 440-nm emission wavelength) using an F-2500 Spectrofluorometer (Hitachi, Tokyo, Japan). The percentage inhibition was calculated using the following formula:% inhibition = 1 − [(fluorescence of test sample)/(fluorescence of control)] × 100.

The effective concentration for 50% inhibition (IC_50_) was obtained by interpolation of the linear regression analysis. All determinations were performed in triplicate, and the results were averaged and compared using the Duncan’s multiple range test (*p* < 0.05).

### 3.8. Statistical Analysis

All values were expressed as mean ± standard deviation and difference between the means of TPC and antioxidant capacity were considered significant at *p* < 0.05 using one way analysis of variance (ANOVA). SPSS package version 18.0 for Windows (IBM Corp, Armonk, NY, USA) was used for the analysis.

## 4. Conclusions

We have successfully developed a precise and accurate HPLC-DAD-ESI-MS method to determine the six major phenolics in *S. formosana*. Based on validation results, the developed method can be useful for detection of phenolics (chlorogenic acid, rutin, isoquercetrin, nictoflorin, astragalin, quercetin) under the specified conditions. The results revealed that methanolic extract and EtOAc fraction from *S. formosana* had effective capacity of scavenging for superoxide radical and correlated with total phenolic content (*R*^2^ = 0.95; *p* < 0.05). The methanol extract of *S. formosana* showed stronger anti-glycation activity, which might be attributed to these abundant phenolics. Therefore, this method can be used for quantification of these bioactive compounds in the herbs and formulations.

## Figures and Tables

**Figure 1 molecules-24-03861-f001:**
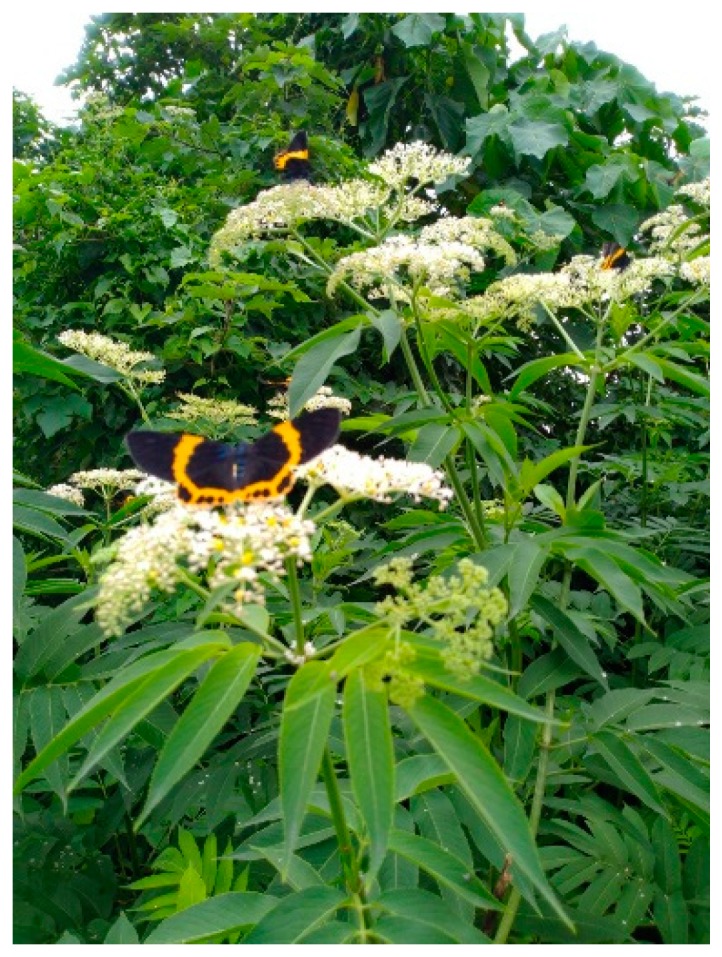
The appearance of *Sambucus formosana*.

**Figure 2 molecules-24-03861-f002:**
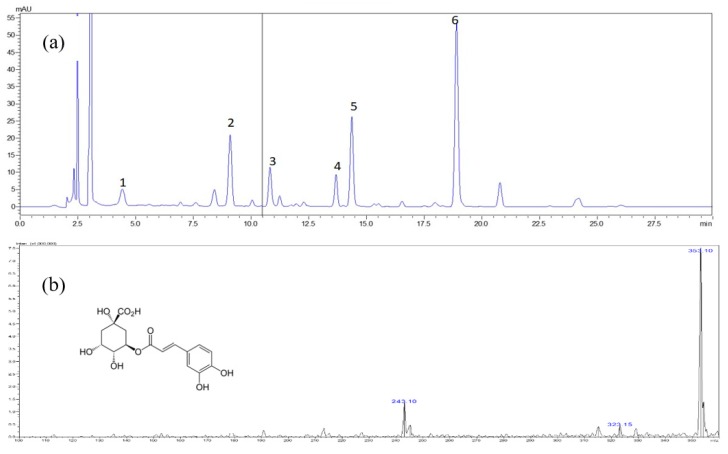
Typical chromatogram of Methanol extract of *S. formosana*. (**a**) High performance liquid chromatography diode array detection (HPLC-DAD) chromatograms obtained at 320 nm.1 = chlorogenic acid, 2 = rutin, 3 = isoquercetrin, 4 = nictoflorin, 5 = astragalin and 6 = quercetin, (**b**) mass spectrometry (MS) spectrum of chlorogenic acid, (**c**) MS spectrum of rutin, (**d**) MS spectrum of isoquercetrin, (**e**) MS spectrum of nictoflorin, (**f**) MS spectrum of astragalin, (**g**) MS spectrum of quercetin.

**Table 1 molecules-24-03861-t001:** Peak assignments of six phenolics compounds.

Peak Number	Retention Time (tR)	[M − H] (*m*/*z*)	Identification
1	4.3	353	Chlorogenic acid
2	9.1	609	Rutin
3	10.7	463	Isoquercetrin
4	13.7	593	Nictoflorin
5	14.3	447	Astragalin
6	18.9	301	Quercetin

**Table 2 molecules-24-03861-t002:** Linear regression equations, coefficients, linear range, precisions and recovery yields, limit of detection and limit of quantification of six phenolics.

Compound	Regression Equation	*R* ^2^	Range (μg/mL)	LOD (μg/mL)	LOQ (μg/mL)	Precision (%)	Recovery (%)
1	y = 6203x − 9406.5	0.997	10–100	1.4	4.3	2.1	86.5
2	y = 9208x − 19934	0.998	10–100	1.0	3.1	1.9	91.2
3	y = 6450x − 20.3	0.997	10–100	1.3	4.0	3.1	93.5
4	y = 5137x + 3696	0.996	10–100	1.2	3.5	2.5	88.5
5	y = 10204x − 5185.2	0.996	10–100	1.5	4.5	1.7	93.1
6	y = 11635x + 24158	0.997	10–100	0.8	2.5	2.7	92.7

**Table 3 molecules-24-03861-t003:** Contents of the six phenolic compounds in extract and fractions from *S. formosana*.

Compound	^a^ MeOH(mg/g)	^a^ EtOAc(mg/g)	^a^ BuOH(mg/g)	^a^ H_2_O (mg/g)
Chlorogenic acid	1.2 ± 0.03	0.9 ± 0.04	0.1 ± 0.03	0.5 ± 0.05
Rutin	2.7 ± 0.05	2.1 ± 0.06	^b^ ND	^b^ ND
Isoquercetrin	2.4 ± 0.02	1.5 ± 0.05	^b^ ND	^b^ ND
Nictoflorin	0.3 ± 0.02	0.4 ± 0.01	^b^ ND	^b^ ND
Astragalin	1.9 ± 0.04	1.1 ± 0.02	^b^ ND	^b^ ND
Quercetin	3.5 ± 0.07	2.2 ± 0.09	0.5 ± 0.02	0.2 ± 0.06

^a^ MeOH = Methanol; EtOAc = ethyl acetate; BuOH = *n*-Butanol; H_2_O = Residual aqueous fraction. ^b^ ND = not detected.

**Table 4 molecules-24-03861-t004:** DPPH (2,2-Diphenyl-1-Picrylhydrazyl) radical scavenging activity and total phenolic content of methanolic extract and fractions from *S. formosana*.

Samples	DPPH Radical Scavenging Activity IC_50_ Value (µg/mL)	Total Phenolic Content (mg Gallic Acid/g Dry Weight)
Methanol extract	97.1 ± 0.2	93.1 ± 0.2
Ethyl acetate fraction	120.5 ± 0.5	69.5 ± 0.2
n-Butanol fraction	520.2 ± 1.5	13.1 ± 0.8
Residual aqueous fraction	590.7 ± 0.4	7.5 ± 0.5

Data values represent the mean ± standard deviation (*n* = 5).

**Table 5 molecules-24-03861-t005:** Anti-glycation activities of Aminoguanidine and extract and fractions from *S. formosana*.

Samples	IC_50_ (μg/mL)
Methanol extract	77.9 ± 0.3
Ethyl acetate fraction	99.6 ± 0.4
*n*-Butanol fraction	>500
Residual aqueous fraction	>500
Aminoguanidine	185.5 ± 0.5

IC_50_ values represent the mean ± SD (*n* = 5).

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
