# Peer review of "HPLC-DAD-ESI-MS Analysis for Simultaneous Quantitation of Phenolics in Taiwan Elderberry and Its Anti-Glycation Activity"

_molecules, 2019, doi:10.3390/molecules24213861_

Round 1
Reviewer 1 Report
I think it's a clear and correct job. It gives a very good impression on a first reading. It is well structured and responds to the main objective set out in line 46: "developing an efficient and validated method for analysis phenolics in S. formosana".
The main problem with the paper comes from the phrase written on lines 44 and 45: "there are no published reports in literature about their compose analysyis, antioxidant activity and on the extraction of phenolics from S. formosana"
This statement is not exactly like that. Because the S. formosana plant, which we accept has been correctly identified by the specialist, has other Latin names, synonyms according to the International Code of Botanical Nomenclature. With that perspective, it should be known that most authors understand that it is synonymous with Sambucus chinensis [see for example the nomenclature and taxonomic comments of Hu Guangwan et al. 2008. Lectotypification of Sambucus chinensis (Caprifoliaceae) and a New Variety from Hunan, China. Novon A Journal for Botanical Nomenclature 18 (1): 61-66 DOI: 10.3417 / 2006055].
If bibliographic surveys are carried out with the keywords "Sambucus chinensis", a good number of published references are retrieved, which should be taken into account as a basis for this paper; They must be mentioned. The results should be interpreted in the light of what is already known about this plant, let's call it Sambucus formosana or call it Sambucus chinensis.
Therefore I suggest that this search be done, and the work be reformulated by making reference - if deemed necessary - to argue why S. formosana and S. chinensis are different or are the same plant.
Author Response
We'd like to appreciate your kind care on our manuscript and all the excellent comments and suggestion. We have modified the manuscript according to your revisions. The revisions are highlight in red color in our revised manuscript. Please see the attachment. Thank you.

Reviewer 2 Report
Please see the attached word document with colours (blue: corrections)

Author Response
We'd like to appreciate your kind care on our manuscript and all the excellent comments and suggestion. We have modified the manuscript point-by-point according to your revisions. The spelling and syntax errors have been checked and corrected, and conclusion section have been expanded base in the results. The revisions are highlight in red color in our revised manuscript. Thank you.

Reviewer 3 Report
Well done, indeed. Therefore, I can most kindly recommend Your valuable manuscript (MS) for publishing in a forthcoming issue of the journal Molecules.
Accept after minor revision.
Please, kindly improve a bit the English language, if You are in position to act in such a way.
In addition to this, the authors are most kindly requested to consider citing of the following references throughout the text of their promising MS (aiming to reinforce the section 2. Results and discussion /phenolics vs. anti-glycation and antioxidant activities, respectively):
Anti-glycation activity - Food Bioscience 2018, Volume 25, Pages 1-7 - Current Pharmaceutical Biotechnology 2017, Volume 18, Issue 15, Pages 1264-1272
Antioxidant activity - Molecules 2018, Volume 23, Issue 8, Article number 1971
Once again my sincere congrats to all authors.
Hopefully, Your MS will be quite well cited (in terms of its hetero-citations) in the time to come.
Last but not least, very best of (research) luck to You all.
Author Response
We'd like to appreciate your kind care on our manuscript and all the excellent comments and suggestion. The reference of 17 and 20 have been added into the revised manuscript. Please see the attachment, thank you.

Reviewer 4 Report
The authors report for the first time the composition of phenolic compounds in Sambucus formosana, traditional herb medicine in Taiwan, and confirm it`s antioxidant and anti-glycation activity.
I recommend this work for publication consideration by the Editorial team of Molecules.
However, there are some concerns needed to be addressed before publication.
It should be interesting if authors add a paragraph about Sambucus chinensis, and to clarify the relationship among S. formosana and S. chinensis in the Introduction section, and also to compare the phenolic compound content in the section Results and Discussion, if possible. The list of references, now numbering 14 references, should be extended.
Section Materials and Methods should be improved. It could be denoted more precisely what is the whole plant, are there some expectations concerning particular plant parts.
What is optimized extract? Was the used HPLC method developed in this study for the first time or followed the previously published procedure? The powder was extracted with 3*120 L methanol?
It should be interesting if authors add a discussion explaining the extraction procedure, the addition of ethylacetat, n-butanol and water, the purpose and expectation.
Some abbreviations, AGE and, BSA, are missing.
In line 127 the authors refer to the reference 16, which is missing in the Reference list.
Author Response
We'd like to appreciate your kind care on our manuscript and all the excellent comments and suggestion. We have modified the manuscript point-by-point according to your revisions. The revisions are highlight in red color in our revised manuscript. Please see the attachment. Thank you.

Round 2
Reviewer 1 Report
The manuscript has been improved from the first draft. It has been very useful to mention S. chinensis/ S. formosana. This clarifies a lot. The Reference list has been changed and now it is more complete and correct.
Reviewer 4 Report
This is a resubmitted manuscript dealing with the composition of phenolic compounds and antioxidant and antiglycation potential of Sambucus formosana.
The manuscript has been significantly improved and now warrants publication in Molecules.
The Introduction section is improved by the addition of relevant references regarding previous work on Sambucus formosana. In previous version, authors were not aware of the previously published reports about the component analysis of Sambucus formosana.
The Reference list which seemed to be too short was expanded, now numbering 23 references, instead of the previous version with just 14 references.
It should be interesting if authors add the discussion concerning the purpose of different fractions analysis.
The English language should be improved.